# The association of diet quality with the mental health of students during their transition to university

**Solomis Solomou \*, Heather Robinson, Guillermo Perez-Algorta**

Faculty of Health and Medicine, Lancaster University, Bailrigg, Lancaster, United Kingdom

\* s.solomou@lancaster.ac.uk

**Data Availability Statement:** All relevant data are within the manuscript and its Supporting Information files. The original datasets can be found at the following URL: https://doi.org/10.17635/lancaster/researchdata/686.

## Abstract

University students are at risk of experiencing mental health and diet quality problems during their transition to university. This study aimed to examine the bidirectional associations between the diet quality and mental health of students during their transition to universities in the UK, and the impact of the transition on diet quality and mental health. The study adopted a cross-sectional design and took place during the first semester of year 2021–2022. Participants were first year undergraduate students at four UK universities, living at student halls. They were invited to participate by completing online surveys, including a diet quality instrument (Short Form Food Frequency Questionnaire), a mental health instrument (Depression Anxiety and Stress Scale) and a quality of transition instrument (College Adjustment Questionnaire). Results showed that a healthy diet was associated with good mental health ($b$ = -3.46, 95% CI [-6.14, -.78]). Unfavourable mental health was associated with having an unhealthy diet ($b$ = -.01, 95% CI [-.02, -.00]). Transition did not moderate the relationship between diet quality and mental health of students, or vice versa ($ps$ > .05). Preliminary analyses showed a significant direct effect of good quality of transition to university on good mental health ($b$ = -1.51, 95% CI [-1.88, -1.13]), but did not have an indirect effect via diet quality; there were no significant direct or indirect effects on diet quality, however larger studies are needed to replicate these preliminary analyses. Our current findings can inform university policies and health education research, and further research is needed to determine whether interventions to improve diet quality at the university level could reduce mental health issues, and whether interventions to support students under stress may lead to healthier dietary habits. Cross-sectional studies cannot determine the directionality of effects, hence longitudinal studies are required to enhance our understanding of the relationships between diet quality and mental health over time.

## Introduction

According to the World Health Organization (WHO), mental health is defined as "a state of mental well-being that enables people to cope with the stresses of life, realise their abilities,

**Funding:** The author(s) received no specific funding for this work.

**Competing interests:** The authors have declared that no competing interests exist.

**Abbreviations:** BMI, Body Mass Index; UCL, University College London; QMUL, Queen Mary University London; KCL, King's College London; SFFFQ, Short Form Food Frequency Questionnaire; DQS, Diet quality score of the SFFFQ; IPAQ, International Physical Activity Questionnaire- Short Form; CAQ, College Adjustment Questionnaire; DASS21, Depression Anxiety and Stress Scale; APSR, Almost Perfect Scale Revised; SPSS, Statistical Package for the Social Sciences; B, Unstandardized beta; SE B, Standard error for the unstandardized beta; CI, Confidence Interval; p, Probability value; R, Correlation coefficient.

learn well and work well, and contribute to their community" [1]. An example of a life stressor is when a transition takes place, where the quality of the transition being defined by the way people respond during a period of change [2]. One such example is when students transition to university.

University students' mental health is a concern and may be negatively affected when transitioning to living on campus [3]. Upon entering university, students may experience stress originating from different sources, such as newfound freedom, lack of social support, increased extracurricular workload, elevated academic expectations, and the pressure associated with making vocational choices [4]. Other factors like financial difficulties, food insecurity, housing instability, perfectionism, and lack of knowledge to shop and cook healthily can further impact their well-being [5]. The difficulties experienced during the transition can lead to the development of mental health issues, such as depression. For example, a meta-analysis of 34 international studies measuring the prevalence of depression in university students showed a substantially higher average prevalence compared to the general population (30% vs 11%) [6,7].

The transition to university life may also affect diet quality. A diet of good quality is perceived as a healthy diet that meets an individual's needs in order to reach optimal health [8]. The term diet quality is also used in literature to define how compliant an individual is to dietary recommendations [9]. Such recommendations include those of the Food Standards Agency in the UK [10].

During the transition to university, students may change their eating habits [11], such as increasing fast food consumption, relying more on take-out meals, and having a reduced intake of fresh food [12,13]. These modifications of diet can lead to a phenomenon called "Freshman 15", which refers to gaining 15 lb (6.8 kg) of weight during the first year of university [14]. A meta-analysis informed by data from multiple countries reported an average weight gain of 1.4 kg over two terms [3]. This weight gain is five times higher compared to changes observed in the general population over a year, and has been observed in at least two thirds of students during their first year of university [15].

The relationships between diet quality and mental health of students, during their transition to universities in the UK, have not been extensively investigated. Moreover, the impact of the quality of the transition on diet and mental health of students at UK universities has not been previously considered by most studies.

This study explored the relationships between diet and mental health during the transition to university, by interrogating data in the context of the stress-diathesis model. The stress-diathesis model can facilitate our understanding of how pre-dispositional factors from various domains can cause susceptibility to psychopathology, and eventually lead to conditions that are sufficient for the development of a disorder, such as a mental health disorder [16]. In the case of our study, we consider the transition to university to act as the stressor for students that may have a vulnerability to following poor quality diets and/or experiencing mental health difficulties, that could lead to negatives outcomes in terms of diet or mental health.

In addition to considering these relationships in the context of the stress-diathesis model, we also recognise the possibility of the transition to university acting as a direct causal factor on diet quality and mental health. For example, financial difficulties and housing instability during this transition period could cause higher levels of worry and/or rumination (indicators of poor mental health), which could be impacting later on diet quality [17,18].

Studies investigating the associations between diet quality and mental health have previously focused on the general population, but studies involving the student population in the UK are scarce. Even in settings outside the UK, relevant studies have not focused on the critical period of transitioning to university. Our study aimed to fill these research gaps, by exploring

the relationships between diet quality and mental health of students, during their transition period to UK based universities.

Our primary research aim was to investigate for any associations of diet quality with the mental health of students and vice versa, during their transition to university, after controlling for gender, perfectionism, exercise, BMI, and university of study. Our secondary aims included investigating whether the quality of the transition moderated the relationship between mental health and diet quality (and vice versa). As part of our secondary research aims, we also investigated whether the transition to university had direct effects on the diet quality and mental health of students. Additionally, we explored whether there were indirect effects of the transition on diet quality through its impact on mental health, as well as on mental health through its influence on diet quality, following a mediational model approach.

## Materials and methods

### Design

The study adopted a cross-sectional design, taking place during the first semester of the academic year 2021–2022 (from 27-SEP-2021 to 17-DEC-2021). Participants were recruited through convenience and snowballing sampling. This study was conducted according to the guidelines laid down in the Declaration of Helsinki [19] and all procedures were approved by the Faculty of Health and Medicine Research Ethics Committee (FHMREC) of Lancaster University, UK (reference FHMREC20128). Written informed consent was obtained online from all participants.

### Participants

Participants were first year undergraduate students at Lancaster University, University College London (UCL), Queen Mary University London (QMUL), and King's College London (KCL) living at student halls (university or private) for the first time. By setting the power of the study to 0.8, and alpha to 0.05, in a model with six predictors in total, we aimed to recruit a minimum of 150 students.

Participants were included if they were of any gender, ethnicity or race; were at least 18 years old; were first year university students for the first time; were studying at the participating universities only. Participants were excluded if they had been staying at their family home during the week prior to survey completion, and/or if they had been fasting over the week prior to survey completion. Eligible students were invited to complete an online survey hosted in REDCap (a secure web application for building and managing online surveys and databases). All electronic data, that were completed online, were anonymous and were stored onto university secure drives.

### Measures

As part of the survey design, we involved members of the public (including university students) and invited them to provide qualitative feedback relevant to the structure and format of the survey. The format of the survey, as well as the phrasing of questions, was adjusted according to the received feedback.

The final questionnaire consisted of general questions that were used to obtain: 1) the socioeconomic information of the participants (age, gender, relationship status, ethnicity, religion, employment status, and food security); 2) lifestyle information (including smoking status, alcohol use, recreational drug use, type of diet followed, knowledge/skills to cook healthily, and any recent life events); and 3) clinical characteristics (including current weight, weight changes, and height).

The effects of the transition to university were measured by the College Adjustment Questionnaire (CAQ) [20]. The questionnaire consists of 14 items, with five items being relevant to educational functioning, five items to relational functioning, and four items to psychological functioning. A general index of adjustment can be computed from these subscales.

The mental health of the participants was monitored by the 21-item depression, anxiety, and stress scale (DASS-21) [21]. The DASS-21 is a self-report scale referring to the week preceding the questionnaire completion. The DASS-21 consists of three sections, each of which consists of seven items measuring depression, anxiety and stress respectively (21 items in total). This tool has been shown to have good construct validity when it was administered to a non-clinical and broadly representative sample of the general adult UK population [22].

Diet quality was measured by the short-form food frequency questionnaire (SFFFQ) [23]. The SFFFQ refers to "a typical week over the past month"; for the purposes of this study, participants were asked to complete the questionnaire in relation to the week preceding the questionnaire completion. The SFFFQ is used to provide a diet quality score (DQS), and was developed through expert knowledge about culturally specific foods, in order to tailor the tool to the UK population. The five dietary components considered as indicators of a healthy diet and used to inform the diet quality score (DQS) were fruit intake, vegetable intake, oily fish intake, fat intake and non-milk extrinsic sugar (NMES) intake. Scores of 1–3 are allocated for each of the above components of the DQS, with a score of 3 corresponding to meeting the UK dietary recommendations for that component. The minimum DQS score is 5 and the maximum score (indicating optimum dietary intake) is 15. The SFFFQ consists of 20 items (multiple choice options) and takes approximately fifteen minutes to complete. Scores of 1–5 were considered as indicating poor diet quality, scores 6–10 as moderate diet quality, and 11–15 as good diet quality. The SFFFQ was validated [23] in comparison to a previously validated and comprehensive food frequency questionnaire (FFQ)- a 217 item FFQ created for the UK Women's cohort study [24]. Participants were also asked to report their diet type (Omnivore, vegetarian, vegan, pescatarian or other).

Physical activity was assessed by the short version of international physical activity questionnaire (IPAQ) [25]. The IPAQ assesses physical activity over the past week, and has been shown to have good reliability and validity, with criterion validity having a median rho of about 0.30 [26].

Perfectionism was assessed by using the Almost Perfect Scale Revised (APSR) [27]. The scale consists of 23 items and has been shown to have good reliability and construct validity [28].

## Statistical analysis

The association of diet quality with mental health, and mental health with diet quality show effect sizes ranging from small to moderate (14). Hence, in a model with six predictors in total, we aimed to recruit 150 students as minimum to obtain 0.8 power, using an alpha of 0.05. This number of participants was also considered suitable to conduct preliminary analyses regarding our secondary questions, with the aim of informing the development of fully powered future studies.

Exploratory analyses were conducted to evaluate the quality of the data, to identify outliers, and to describe participant characteristics in terms of demographic and clinical information. To respond to the primary research questions, hierarchical regression analyses were conducted, with mental health or diet quality as the dependent variables, and mental health or diet quality, gender, perfectionism, exercise, BMI, and university of study as predictors.

For the moderation analysis we centred our three main variables using sample means (mental health, diet quality and transition to university), including interactive effects accordingly (transition*mental health or transition*diet quality).

Mediation analyses were also performed, following approaches that generate 95% confidence intervals using bootstrapping [29]. To run the analyses, the software SPSS (Statistical Package for the Social Sciences) was used [30].

## Results

### Sample characteristics

The final sample consisted of n = 166 students. The sample average age was 19.4 years (SD = 2.9 years), 71% female, of white ethnic background (59%) and "single" (77%). The sample characteristics are summarised in Table 1.

In terms of diet (Table 2), most students (68%) reported to be omnivore (i.e. eating various types of food, including meat, vegetables and other animal products), and 76.5% felt they had knowledge and skills to cook healthily. Among respondents, 79% did not experience any food security issues during the 12 months before survey completion while 87% did not report any weight changes since starting university. The mean BMI was 23 kg/m$^2$ (SD = 4.9 kg/m$^2$), a result within the healthy adult BMI range of 18.5–24.9 [31].

The mean score of the short form food frequency questionnaire (SFFFQ) was 9.8 (SD = 1.7) out of a maximum score of 15 (Fig 1, panel A), which indicates a moderate-good diet quality. In terms of physical activity, the mean value for metabolic minutes per week was 3590 (SD = 2898) (Fig 1, panel B), which indicates a high level of physical activity. Moreover, 51.8% of the students were under the category of high level of physical activity.

Forty-three percent of the sample reported having experienced stressful events since starting university (Table 3, Fig 2), with financial difficulties being the most frequently reported stressful event. As shown in Table 3, the group that experienced stressful events since starting university exhibited poorer mental health quality compared to the group that did not report stressful events. Overall, this student sample had a reasonably good quality of transition, with lower psychological functioning scores compared to the educational functioning and relational functioning scores (Table 3, Fig 3).

Regarding our primary research questions, the results showed that diet quality was a significant predictor of mental health, ($b$ = -3.46, 95% CI [-6.14, -.78]). Diet quality explained 4% of the variability in overall mental health scores, in a model where all predictors together explained 11% of variance ($R^2$ = .11, F(6, 137) = 2.40, p ≤ .05) (Table 4).

When the model was tested in the opposite direction (research aim 2), mental health was a significant predictor of diet quality ($b$ = -.01, 95% CI [-.02, -.00]). Mental health explained 4% of variance on diet quality, in a model that explained 11% in total ($R^2$ = .11, F(7, 136) = 2.46, p ≤ .05) (Table 5).

Tables 6 and 7 present results for our secondary research questions. With mental health being the dependent variable and all predictors in the model, the interaction effect of the transition and diet quality was not statistically significant ($p$ = .27). Diet quality remained as a statistically significant predictor ($b$ = -3.13, 95% CI [-5.33, -.93]) and contributed 4% of the total variance explained. Transition scores were also statistically significant ($b$ = -1.51, 95% CI [-1.88, -1.13]) and contributed 31% of the variance. Increases on the quality of both diet and transition were associated with reductions in mental health scores.

In a model with diet quality as the dependent variable the interaction effect of the transition and mental health was not statistically significant ($p$ = .17). Mental health was a significant predictor ($b$ = -.02, 95% CI [-.03, -.00]), explaining 6% of the total variance.

When testing whether the quality of the transition to university had an indirect effect through diet quality on mental health (Fig 4), this indirect effect was not statistically significant (95% bootstrap CI, -.08 to .05). Only the direct effect of the transition to university was found

**Table 1. Demographic information.**

| Variables | |
|---|---|
| **Age (years)** | |
| *n (missing n)* | 162 (4) |
| *Mean (SD)* | 19.4 (2.9) |
| *Range* | 18–41 |
| **Gender, *n (%)*** | |
| *Female* | 118 (71) |
| *Male* | 44 (27) |
| *Other* | 4 (2) |
| **University, *n (%)*** | |
| *Lancaster University* | 55 (33) |
| *UCL* | 13 (8) |
| *QMUL* | 57 (34) |
| *KCL* | 37 (22) |
| *Not known* | 4 (2) |
| **University Location, *n*** | |
| Urban (London) | 107 (65) |
| Rural (Lancaster) | 55 (31) |
| Not known | 4 (2) |
| **Ethnic background, n (%)** | |
| *White* | 98 (59) |
| *Asian or Asian British* | 24 (15) |
| *Chinese* | 14 (8) |
| *Other (Black, mixed, other ethnic group)* | 30 (18) |
| **Relationship status, n (%)** | |
| *Single* | 127 (77) |
| *In a relationship* | 35 (21) |
| *Other* | 4 (2) |
| **Student status, n (%)** | |
| *Home Student* | 115 (69) |
| *Overseas Student* | 50 (30) |
| *Not known* | 1 (1) |
| **Smoking status, n (%)** | |
| *Current smoker* | 19 (11) |
| *Ex-smoker* | 5 (3) |
| *Never smoked > 100 cigarettes* | 142 (86) |
| **Alcohol consumption (last week), n (%)** | |
| *Rarely/no alcohol* | 82 (49) |
| *<14 units* | 55 (33) |
| *14–21 units* | 21 (13) |
| *>21 units* | 8 (5) |
| Use of recreational drugs, n (%) | |
| *Yes* | 11 (7) |
| *No* | 155 (93) |

University College London (UCL), Queen Mary University London (QMUL), King's College London (KCL), Standard Deviation (SD).

**Table 2. Information about diet, weight and physical activity.**

| Variables | |
|---|---|
| **Food insecure (last 12 months), n (%)** | |
| Often | 5 (3) |
| Sometimes | 30 (18) |
| Never | 131 (79) |
| **Type of diet followed, n (%)** | |
| Omnivore | 113 (68) |
| Vegetarian | 29 (18) |
| Vegan | 5 (3) |
| Pescatarian | 7 (4) |
| Other (Halal, Kosher) | 12 (7) |
| **Subjective knowledge/skills for healthy cooking n (%)** | |
| Yes | 127 (76.5) |
| No | 39 (23.5) |
| **Weight (kg)** | |
| n (missing n) | 162 (4) |
| Mean (SD) | 65.1 (15.7) |
| Range | 39.1–131.0 |
| **Weight change since starting university** | |
| No change | 87 (52) |
| n (%) | 27 (16) |
| Weight gain (kg) | 3.8 (3.0) |
| n (%) | 50 (30) |
| mean (SD) | 3.4 (3.6) |
| Weight loss (kg) | 2 (1) |
| n (%) | |
| mean (SD) | |
| Not known | |
| n (%) | |
| **BMI (kg/m$^2$)** | |
| n (missing n) | 156 (10) |
| Mean (SD) | 23.0 (4.9) |
| Range | 14.7–41.0 |
| **Diet quality score (SFFFQ)** | |
| n (missing n) | 166 (0) |
| Mean (SD) | 9.8 (1.7) |
| Range | 5–14 |
| **Physical activity n (%)** | |
| Low | 13 (7.8) |
| Moderate | 61 (36.7) |
| High | 86 (51.8) |
| Not known | 6 (3.6) |
| Metabolic minutes per week | 160 (6) |
| n (missing n) | 3590 (2898) |
| Mean (SD) | 0–14238 |
| Range | |

Body Mass Index (BMI), Standard Deviation (SD), short form food frequency questionnaire (SFFFQ).

to be statistically different ($t(161)$ = -8.66; $p \leq .01$), meaning that with the level of diet quality being held constant, a better quality of transition by one unit lead to a better mental health score by 1.53 units (Table 8).

*Panel A. Distribution of the diet quality scores of the SFFFQ questionnaire*   *Panel B. Distribution of the metabolic minutes per week of the IPAQ questionnaire*

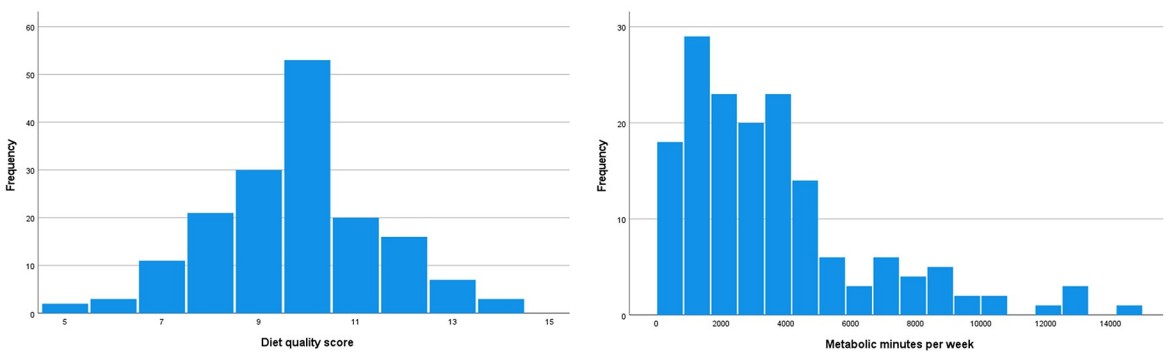

**Fig 1. Diet quality and exercise score distributions.**

**Table 3. Information about mental health and adjustment to university.**

| Demographics/variables | All | Stressful events | No stressful events | t-test |
|---|---|---|---|---|
| Stressful life events since starting university n (%) | 166 (100) | 72 (43) | 94 (57) | N/A |
| **Adjustment to university score (CAQ)** | | | | |
| Educational functioning | 17.7 (3.7) | 17.4 (4.0) | 18.0 (3.4) | t(164) = 1.062, |
| *Mean (SD)* | 5–25 | 5–25 | 9–25 | p = 0.290 |
| *Range* | 15.5 (5.3) | 14.8 (5.3) | 15.9 (5.3) | t(164) = 1.327, |
| Relational functioning | 5–25 | 5–25 | 5–25 | p = 0.186 |
| *Mean (SD)* | 13.4 (4.1) | 11.9 (4.0) | 14.5 (3.7) | t(164) = 4.387, |
| *Range* | 4–20 | 5–20 | 4–20 | p < 0.001 |
| Psychological functioning | | | | |
| *Mean (SD)* | | | | |
| *Range* | | | | |
| *Mental health score (DASS21)* | | | | |
| Depression | 14.7 (11.5) | 17.3 (11.4) | 12.7 (11.2) | t(164) = 2.586, |
| *Mean (SD)* | 0–42 | 0–42 | 0–42 | p = 0.011 |
| *Range* | 13.3 (10.1) | 16.9 (9.9) | 10.5 (9.5) | t(164) = 4.219, |
| Anxiety | 0–42 | 0–42 | 0–42 | p < 0.001 |
| *Mean (SD)* | 15.8 (10.7) | 19.6 (10.7) | 12.9 (9.9) | t(164) = 4.184, |
| *Range* | 0–42 | 0–42 | 0–40 | p < 0.001 |
| Stress | 43.8 (28.7) | 53.8 (27.8) | 36.1 (27.1) | t(164) = 4.115, |
| *Mean (SD)* | 0–126 | 0–126 | 0–118 | p < 0.001 |
| *Range* | | | | |
| Total DASS21 score | | | | |
| *Mean (SD)* | | | | |
| *Range* | | | | |
| **Perfectionism score (APSR)** | | | | |
| *n (missing n)* | 162 (4) | 71 (1) | 91 (3) | t(160) = 1.230 |
| *Mean (SD)* | 32.0 (7.4) | 32.8 (7.7) | 31.3 (7.1) | p = 0.220 |
| *Range* | 9–42 | | | |

College Adjustment Questionnaire (CAQ), Depression Anxiety and Stress Scale (DASS21), Almost Perfect Scale Revised (APSR), Standard Deviation (SD).

*Panel A. Distribution of the depression scores of the DASS21 questionnaire*

*Panel B. Distribution of the anxiety scores of the DASS21 questionnaire*

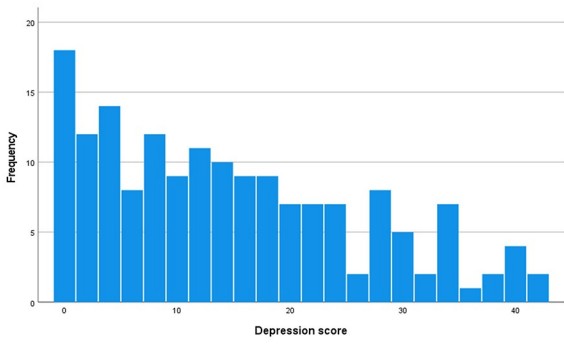

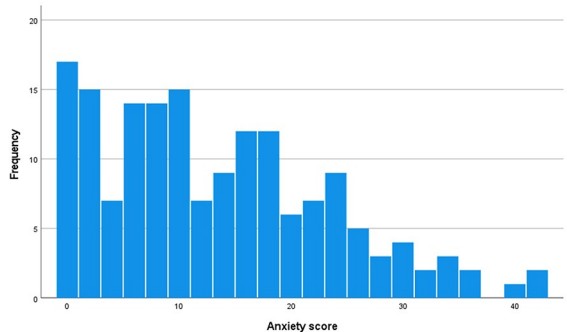

*Panel C. Distribution of the stress scores of the DASS21 questionnaire*

*Panel D. Distribution of the total scores of the DASS21 questionnaire*

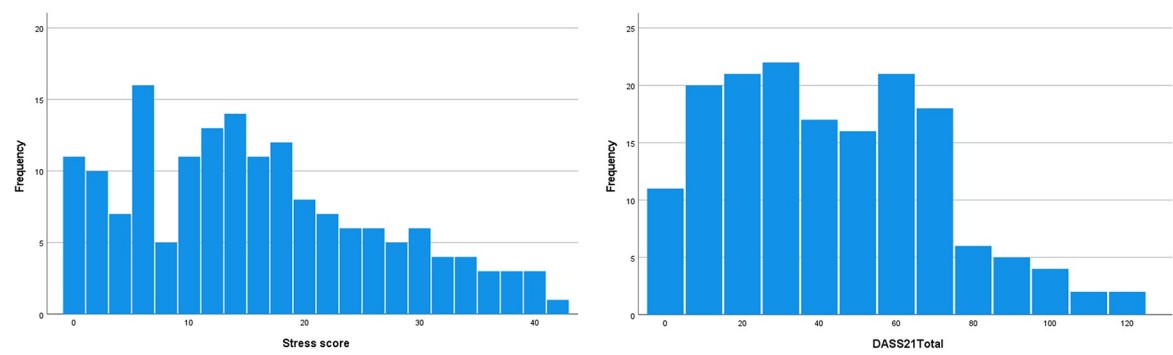

**Fig 2. Distribution of the depression, anxiety, stress and total scores of the DASS21 questionnaire.** Depression Anxiety and Stress Scale (DASS21).

In a second model examining whether the quality of the transition to university had an indirect effect through mental health on diet quality (Fig 5), this indirect effect did not reach statistical significance (95% bootstrap CI, -.00 to .04). Additionally, no significant results were observed for the direct effect of transition on diet quality (*p* = .38) (Table 9).

## Discussion

This was one of the first studies to investigate the associations between diet quality and mental health of students during the transition to university at UK universities. The study results showed that having a good diet quality during the transition to university was associated with better mental health. The effect sizes were small and should be interpreted with caution.

The study results are in line with the findings of our systematic literature review [32], which found that good diet quality of students was associated with better mental health in terms of depression, anxiety, stress and overall general mental well-being, with the effect sizes being small-moderate. Our results are also in line with previous research with both students transitioning to university [28] and the general population [29] where better quality diet has been found to be associated with better mental health. A particular parallel is observed with a

*Panel A. Distribution of the educational functioning scores of the CAQ questionnaire*

*Panel B. Distribution of the relational functioning scores of the CAQ questionnaire*

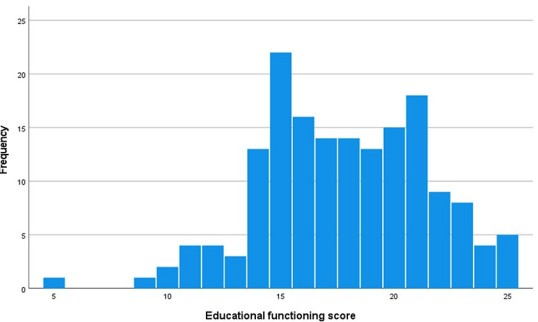

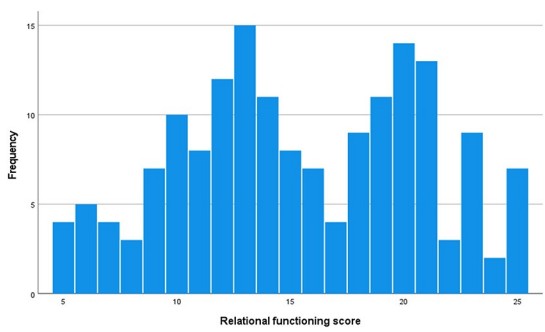

*Panel C. Distribution of the psychological functioning scores of the CAQ questionnaire*

*Panel D. Distribution of the standards scores of the APSR questionnaire*

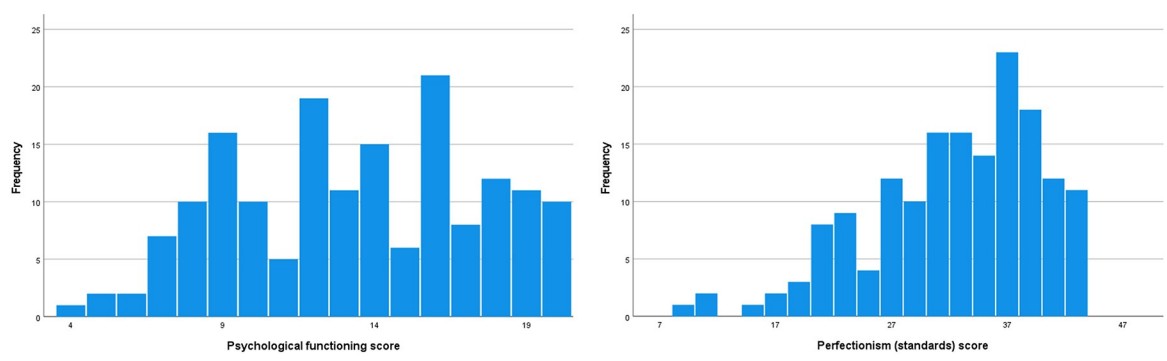

**Fig 3. Distribution of the educational, relational and psychological functioning scores of the CAQ questionnaire, and of the standards scores of the APSR questionnaire.** College Adjustment Questionnaire (CAQ), Almost Perfect Scale Revised (APSR).

previous study of students transitioning to a UK-based university, which found that the group with the worst diet quality had higher levels of depression and anxiety compared to groups with better diet quality [33]. Previous studies involving the general population [34] have also found that good quality diets were associated with good mental health.

We also found statistically significant differences in the mean diet quality scores depending on diet type, although these results should be interpreted with caution due to the small sample size of some diet type groups. The pescatarian diet was associated with a statistically significant higher diet quality mean score than the omnivore, vegetarian diet and other (Halal and Kosher) diets. This could imply that the mechanisms responsible for the associations observed in our study could be driven by the presence of elements of Mediterranean diet. It is well known that following healthy diets, like the Mediterranean diet, reduces the risk of depression in the general population [35]. These diets reduce inflammation, which could play a contributory role in psychiatric disorders [36,37]. The Mediterranean diet is rich in fruits and vegetables, whole grains, nuts and legumes, and olive oil, but also seafood.

Other possible explanations for the observed associations might include vulnerabilities students face during the transition period. For example, the lack of regularity and the loss of home rituals may mean that regular home-made food is replaced by irregular and unhealthy

**Table 4. Sequential regression of variables (including diet quality scores) on the overall mental health (DASS-21).**

| Variables | DASS21 score | Gender 0 (female) | Gender 1 (male) | Gender 2 (other) | Perfectionism | Exercise | BMI (kg/m²) | University Location | DQS score | B | SE B | 95% CI for B | sr² (incremental) |
|---|---|---|---|---|---|---|---|---|---|---|---|---|---|
| Gender 0 (female) | .14* | | | | | | | | | | | | |
| Gender 1 (male) | -.20** | -.93** | | | | | | | | -12.80 | 5.50 | -23.66, -1.93 | |
| Gender 2 (other) | .15* | -.26** | -.10 | | | | | | | 17.02 | 14.07 | -10.81, 44.84 | .06* |
| Perfectionism (APSR) | .15* | .14* | -.20** | .13 | | | | | | .37 | .31 | -.25, .99 | .01 |
| Exercise | -.02 | -.16* | .14* | .07 | .05 | | | | | .00 | .00 | -.00, .00 | .00 |
| BMI | .01 | -.09 | .10 | -.02 | -.02 | -.10 | | | | .13 | .48 | -.81, 1.07 | .01 |
| University Location | .03 | .23** | -.21** | -.07 | .08 | -.20** | -.06 | | | .37 | 5.12 | -9.75, 10.49 | .00 |
| DQS score | -.19** | .15* | -.13 | -.07 | .01 | .18* | -.07 | .06 | | -3.46 | 1.35 | -6.14, -.78 | .04** |
| Intercept | | | | | | | | | | 64.60 | 20.63 | | |
| Means | 44.24 | .71 | .26 | .03 | 31.48 | 3353.69 | 23.06 | .69 | 9.73 | | | | |
| Standard deviations | 28.06 | .46 | .44 | .16 | 7.46 | 2691.68 | 4.85 | .47 | 1.74 | | | | |
| | | | | | | | | | | | | | R² = .11 Adjusted R² = .06 R = .33 |

Depression Anxiety and Stress Scale (DASS21), Almost Perfect Scale Revised (APSR), Diet Quality Score of the Short Form Food Frequency Questionnaire (DQS), Body Mass Index (BMI).

Unstandardized beta (B), standard error for the unstandardized beta (SE B), CI (confidence interval), probability value (p), correlation coefficient (R) is the correlation between the observed values of the response variable and the predicted values of the response variable made by the model, coefficient of determination ($R^2$) is the proportion of the variance in the response variable that can be explained by the predictor variables in the regression model.

Semi-partial correlations squared ($sr^2$) represent the percentage of variance in the dependent variable that is uniquely associated with each independent variable.

*$p \leq .05$

**$p \leq .01$

fast-food. In addition, when transitioning to university, students may face financial difficulties. In fact, financial difficulties since starting university was the most frequent stressful event reported by our study participants. Financial difficulties can lead to food insecurity, resulting in students relying more on cheaper unhealthy food- a negative change compared to the food they used to have at home. The effects of these changes could be contributing towards a deterioration of mental health, as students perceive a worsening of their diet and a loss of their usual home dietary routines [38].

Moreover, the fact that students now have to rely on their own cooking means that the quality of the foods they prepare may not be of the same high standard as the food they used to have at home- in cases where they lack the knowledge to cook healthily. Our findings showed that students reporting good knowledge of cooking healthily had better diet quality than those that had no knowledge. There may also be an impact on mental health, as when being unable to cook to the same standard as home food students may experience negative feelings, such as inadequacy and inability to cope by themselves [39].

Our results also showed that there was a statistically significant association of good mental health with good diet quality, however the association was weaker compared to the association

**Table 5. Sequential regression of variables (including overall mental health scores) on the diet quality (DQS) score.**

| Variables | DQS score | Gender 0 (female) | Gender 1 (male) | Gender 2 (other) | Perfectionism | Exercise | BMI (kg/m²) | University Location | DASS21 score | B | SE B | 95% CI for B | sr² (incremental) |
|---|---|---|---|---|---|---|---|---|---|---|---|---|---|
| Gender 0 (female) | .15* | | | | | | | | | | | | |
| Gender 1 (male) | -.13 | -.93** | | | | | | | | -.75* | .34 | -1.42, -.07 | |
| Gender 2 (other) | -.07 | -.26** | -.10 | | | | | | | -.71 | .87 | -2.44, 1.01 | .02 |
| Perfectionism (APSR) | .01 | .14* | -.20** | .13 | | | | | | -.00 | .02 | -.04, .04 | .00 |
| Exercise | .18* | -.16* | .14* | .07 | .05 | | | | | .00 | .00 | .00, .00 | .04** |
| BMI | -.07 | -.09 | .10 | -.02 | -.02 | -.10 | | | | -.01 | .03 | -.07, .05 | .00 |
| University Location | .06 | .23** | -.21** | -.07 | .08 | -.20** | -.06 | | | .24 | .32 | -.39, .86 | .00 |
| DASS21 score | -.19* | .14* | -.20** | .15* | .15* | -.02 | .01 | .03 | | -.01 | .01 | -.02, -.00 | .04** |
| Intercept | | | | | | | | | | 10.15 | 1.00 | | |
| Means | 9.73 | .71 | .26 | .03 | 31.48 | 3353.69 | 23.06 | .69 | 44.24 | | | | |
| Standard deviations | 1.74 | .46 | .44 | .16 | 7.46 | 2691.68 | 4.85 | .47 | 28.06 | | | | |
| | | | | | | | | | | | | | R² = .11 Adjusted R² = .07, R = .34* |

Depression Anxiety and Stress Scale (DASS21), Almost Perfect Scale Revised (APSR), Diet Quality Score of the Short Form Food Frequency Questionnaire (DQS), Body Mass Index (BMI).

Unstandardized beta (B), standard error for the unstandardized beta (SE B), CI (confidence interval), probability value (p), correlation coefficient (R) is the correlation between the observed values of the response variable and the predicted values of the response variable made by the model, coefficient of determination (R²) is the proportion of the variance in the response variable that can be explained by the predictor variables in the regression model.

Semi-partial correlations squared (sr²) represent the percentage of variance in the dependent variable that is uniquely associated with each independent variable.

*p ≤ .05

**p ≤ .01.

of diet quality with mental health. It might be the case that certain mental health difficulties may be more strongly associated with diet quality than others, for example anxiety and stress [32]. The effect sizes were small and should be interpreted with caution.

Contrary to expectations based on the stress-diathesis model, our study found no statistically significant moderational effect. This could be attributed to the participants' generally good diet quality and mental health scores, which may have limited variability, making it difficult to detect such effects. Additionally, the study might have lacked the statistical power necessary to identify small effect sizes. Future studies with larger samples are needed to replicate these findings.

Apart from acting as a moderator, we also considered the possibility of transition having a causal effect. In view of this, we ran a mediational model to investigate for any direct and/or indirect effects via diet quality or mental health. We found that quality of the transition to university had a significant direct effect on mental health of students, but not on diet quality. Additionally, our study found no statistically significant indirect effects. The reasons behind the lack of significant findings could be due to the limited variability and small size of the sample. Larger samples with participants having a broader variability of diet quality may provide

**Table 6. Sequential regression of variables (including multiplicative interactive effects of transition and diet quality) on the overall mental health (DASS-21).**

| Variables | B | SE B | 95% CI for B | β | sr² (incremental) |
|---|---|---|---|---|---|
| Gender 0 (female) | | | | | |
| Gender 1 (male) | -6.73 | 4.55 | -15.73, 2.27 | -.11 | |
| Gender 2 (other) | 13.35 | 11.48 | -9.35, 36.05 | .08 | .06* |
| Perfectionism (APSR) | .51 | .26 | -.00, 1.01 | .13 | .01 |
| Exercise | .00 | .00 | .00, .00 | .13 | .00 |
| BMI (kg/m²) | -.20 | .39 | -.98, .57 | -.04 | .00 |
| University Location | -.85 | 4.19 | -9.14, 7.44 | -.01 | .00 |
| CAQ score | -1.51** | .19 | -1.88, -1.13 | -.56 | .31** |
| DQS score | -3.13** | 1.11 | -5.33, -.93 | -.19 | .04** |
| CAQ*DQS | .13 | .12 | -.10, .36 | .08 | .01 |
| | | | | | R² = .42 Adjusted R² = .38 R = .65** |

Body Mass Index (BMI), Depression Anxiety and Stress Scale (DASS21), Almost Perfect Scale Revised (APSR), College Adjustment Questionnaire (CAQ), Diet Quality Score of the Short Form Food Frequency Questionnaire (DQS), CAQ*DQS multiplicative effect of transition and diet quality Unstandardized beta (B), standard error for the unstandardized beta (SE B), Confidence Intervals (CI), standardized beta (β), probability value (p), correlation coefficient (R) is the correlation between the observed values of the response variable and the predicted values of the response variable made by the model, coefficient of determination (R²⁾ is the proportion of the variance in the response variable that can be explained by the predictor variables in the regression model.

Semi-partial correlations squared (sr²) represent the percentage of variance in the dependent variable that is uniquely associated with each independent variable.

*p ≤ .05

**p ≤ .01.

further insights. Moreover, studies involving participants with established mental health difficulties may enhance our understanding as to whether the transition has any indirect effects on mental health via diet quality in this student group.

Future studies could also explore new potential moderators and mediators. These include knowledge to cook and shop healthily, food security, weight changes, access to traditional food as moderators, as well as biological mechanisms as candidate mediators. We recommend that future studies use longitudinal designs, enabling researchers to determine the direction of any detected associations. Qualitative studies are also warranted to explore the daily experience of students during this particular period of time.

Apart from having statistical significance, our findings have practical significance and several important implications. For example, the findings highlight the need to consider strategies at the university level to raise awareness of the associations between diet and mental health. The findings also emphasise the fact that universities need to review how they operate in order to promote better diet quality and mental health of students during their university experience, and especially during the transition to university.

Considering the fact that food insecurity may be a factor affecting the diet quality of students, food assistance services may be helpful for food insecure students [5]. The quality of food offered as part of food assistance would also be relevant, as diet quality may only improve if fresh and healthy options are offered as part of the food assistance, rather than options that may be low in terms of nutritional quality [40].

The study findings suggest that good knowledge to cook healthily is associated with a better diet quality of students. This means that there may be scope to target and improve the food

**Table 7. Sequential regression of variables ((including multiplicative interactive effects of transition and mental health) on the diet quality (DQS) score.**

| Variables | B | SE B | 95% CI for B | β | sr² (incremental) |
|---|---|---|---|---|---|
| Gender 0 (female) | | | | | |
| Gender 1 (male) | -.64 | .34 | -1.31, .041 | -.16 | |
| Gender 2 (other) | -.58 | .87 | -2.30, 1.14 | -.06 | .02 |
| Perfectionism (APSR) | .00 | .02 | -.04, .04 | .01 | .00 |
| Exercise | .00 | .00 | .00, .00 | .24 | .04* |
| BMI (kg/m²) | -.01 | .03 | -.07, .04 | -.04 | .00 |
| University Location | .18 | .32 | -.45, .80 | .05 | .00 |
| CAQ score | -.02 | .02 | -.06, .01 | -.15 | .00 |
| DASS-21 score | -.02** | .01 | -.03, -.00 | -.30 | .06** |
| CAQ*DASS-21 | .00 | .00 | .00, .00 | .11 | .01 |
| | | | | | $R^2$ = .14 Adjusted $R^2$ = .08 R = .37* |

Body Mass Index (BMI), Depression Anxiety and Stress Scale (DASS21), Almost Perfect Scale Revised (APSR), College Adjustment Questionnaire (CAQ), Diet Quality Score of the Short Form Food Frequency Questionnaire (DQS), CAQ*DASS-21 multiplicative effect of transition and mental health Unstandardized beta (B), standard error for the unstandardized beta (SE B), confidence intervals (CI), standardized beta (β), probability value (p), correlation coefficient (R) is the correlation between the observed values of the response variable and the predicted values of the response variable made by the model, coefficient of determination ($R^{2)}$) is the proportion of the variance in the response variable that can be explained by the predictor variables in the regression model.

Semi-partial correlations squared (sr²) represent the percentage of variance in the dependent variable that is uniquely associated with each independent variable.

*p ≤ .05

**p a≤ .01.

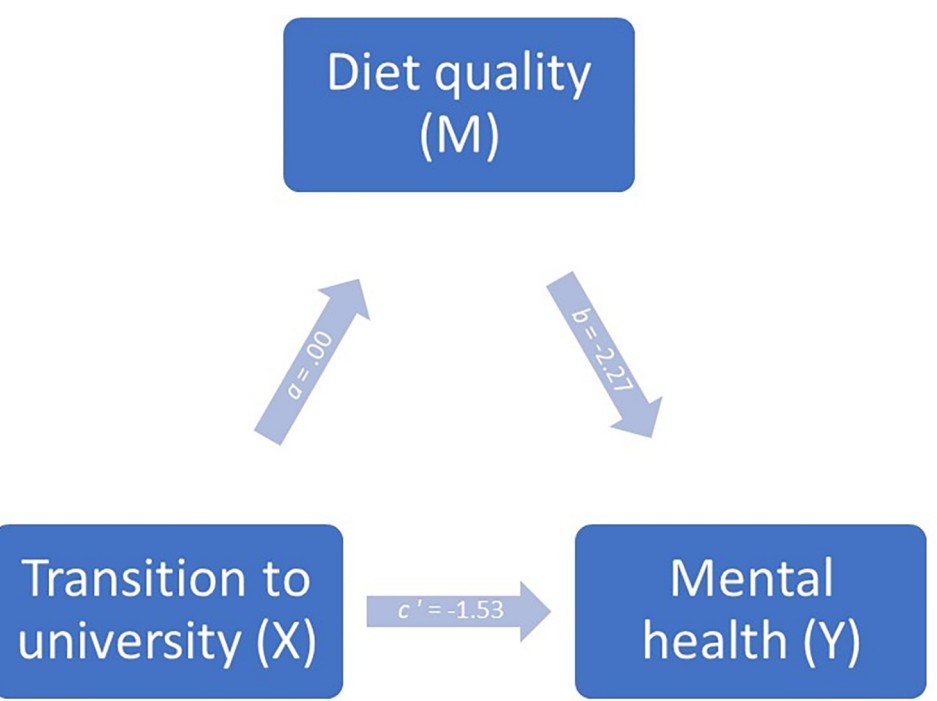

**Fig 4. Simple mediation model for the influence of the transition to university and diet quality on the mental health of university students, in the form of a statistical diagram.**

**Table 8. Model coefficients for the influence of transition to university on mental health.**

| Consequent | | | | | | | | |
|---|---|---|---|---|---|---|---|---|
| | *M* (Diet quality) | | | | *Y* (Mental health) | | | |
| Antecedent | | Coeff. | *SE* | *p* | | Coeff. | *SE* | *p* |
| *X* (Transition) | *a* | .00 | .01 | .72 | *c′ b* | -1.53 | .18 | < .001 |
| *M* (Diet quality) | | — | — | — | | -2.27 | 1.06 | < .05 |
| constant | *iM* | 9.53 | .62 | < .001 | *iY* | 137.33 | 13.19 | < .001 |
| | | | $R^2 = .00$ | | | | $R^2 = .33$ | |
| | | $F(1, 164) = .13, p = .72$ | | | | $F(2, 163) = 40.29, p < .001$ | | |

Standard deviation (SE), Probability value (p), Coefficient (coeff.), correlation coefficient (R), Fvalue (F).

**Table 9. Model coefficients for the influence of transition to university on diet quality.**

| Consequent | | | | | | | | |
|---|---|---|---|---|---|---|---|---|
| | *M* (Mental health) | | | | *Y* (Diet quality) | | | |
| Antecedent | | Coeff. | *SE* | *p* | | Coeff. | *SE* | *p* |
| *X* (Transition) | *a* | -1.54 | .18 | < .001 — | *c′ b* | -.01 | .02 | .38 |
| *M* (Mental health) | | — | — | | | -.01 | .01 | < .05 |
| constant | *iM* | 115.74 | 8.55 | < .001 | *iY* | 10.91 | .89 | < .001 |
| | | | $R^2 = .31$ | | | | $R^2 = .03$ | |
| | | $F(1, 164) = 74.42, p = < .001$ | | | | $F(2, 163) = 2.34, p = .10$ | | |

Standard deviation (SE), Probability value (p), Coefficient (coeff.), correlation coefficient (R), Fvalue (F).

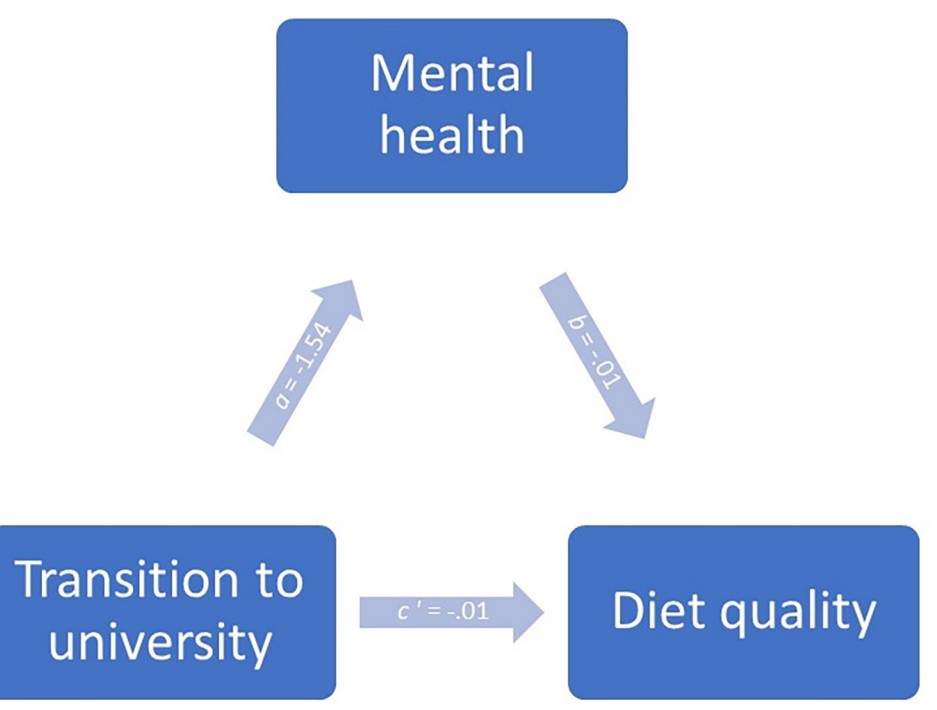

**Fig 5. Simple mediation model for the influence of the transition to university and mental health on the diet quality of university students, in the form of a statistical diagram.**

literacy of university students; this may increase the confidence of students to cook healthily by themselves rather than relying on take away foods of low nutritional value, thus leading to a better diet quality. Food literacy would not just include cooking skills, but also education relevant to planning, managing, budgeting, selecting and preparing food [5,41]. Interventions that could be considered include nutrition education lectures, cooking demonstrations, and discussion [42], individualised dietary counselling meetings with a clinical nutritionist [43], dietary advice via newsletters [44].

In terms of interventions targeting the mental health of students, these could include traditional counselling and technology-delivered psychological interventions, (including online cognitive behavioural therapy) [45], mindfulness-based interventions [46], educational/personalised mail feedback interventions to provide students with feedback about their symptoms [47], recreation programmes (such as meditation and exercise), as well as psychoeducation interventions to provide information relevant to stress, coping, and ways to relax [48].

Our study did not collect information about the food environment and food purchasing at the university campuses, as this was not the study scope. However, previous studies have reported that a higher frequency of food and beverage purchasing on campuses was associated with the consumption of energy-dense and nutrient-poor foods [49,50]. There may be scope for relevant university policies to be developed. For example, universities may need to review what economic and planning decisions they make in terms of which franchises they allow onto campuses, and influence what is sold in the university shops. Additionally, there could be consideration for exclusion zones around the university campuses for franchises and shops that offer unhealthy foods. Development of such policies would allow monitoring the types of food stores allowed on campuses, as well as around campuses; this may limit the availability of fast foods and maximise the availability of healthy options at cheaper prices [51].

There are a number of strengths to this study. For example, the study had a strong theoretical foundation; it did not adopt a narrow biomedical approach, and instead used a stress diathesis model of biopsychosocial factors to investigate and analyse exposure and outcomes, taking into consideration relevant confounding variables. The study consisted of more participants (n = 166) than the minimum necessary (n = 150), meaning it was adequately powered to answer the primary aims.

The study sample was representative in terms of sociodemographic composition, as this was comparable with that of the overall undergraduate student population in the UK. The sample was also balanced in terms of geographical location, in terms of rural vs urban settings. The study sample included participants of various ethnic backgrounds (White, Asian or Asian British, Chinese, Black, Mixed, other), but the majority of the participants were of White ethnic background (59%). This is in line with the demographics for UK universities, as in academic year 2020–2021 students of White ethnicity accounted for a majority of 74% of all UK domiciled enrolments [52]. The sample was also in line with the ratio of home to international students at UK universities.

There were also some limitations to this study. For example, the study was cross-sectional therefore the directionality of effects could not be determined. The study only addressed depression, anxiety and stress, however there are other mental health issues such as eating disorders that often develop around the age of moving to university. Hence, we cannot rule out the possibility that some students with eating disorders may have taken part in our study. Another issue that was not addressed by our study is that relationships between diet quality and mental health during the transition to university could also be partly explained by biological mechanisms, such as inflammation and dysregulated hypothalamic pituitary adrenal (HPA) axis.

The study relied on self-reported measures for both diet quality and mental health, which may have introduced biases. For example, it has previously been shown that that individuals underreport their weight in self-reports, and that individuals with higher body mass index (BMI) underestimate their weight more than those with lower BMI [53]. Future studies may consider assessing participants in person in order to avoid such biases.

Cross-sectional studies, such as our study, limit the ability to draw causal inferences between diet quality and mental health. This highlights the need for the design of longitudinal studies in the future, in order to better capture the dynamics of the relationships between diet quality and mental health over time.

Previous studies have suggested that there may be biological mechanisms affecting the relationships between diet quality and mental health, such as inflammation and dysregulated hypothalamic pituitary adrenal (HPA) axis [54]. Our study did not explore biological mechanisms, meaning there is a need for future longitudinal studies to take biological factors into consideration.

Even though the sample size was adequate to power the primary analyses, it may have been insufficient to detect smaller effects, particularly in the moderation and mediation analyses. Additionally, even though the study was powered to take five possible confounding variables into consideration, there may be other potential confounding variables that could influence the relationship between diet and mental health, such as socioeconomic status, pre-existing mental health conditions, financial distress, alcohol and recreational drug use. Studies with larger samples are needed in order to take additional possible confounding factors into consideration.

Issues at the time of the study included lockdowns and restrictions relevant to COVID-19, as well as a cost of living crisis. We cannot exclude the possibility that these factors may have made the transition for students more difficult than expected, which may affect the generalisability of the results for periods without COVID-19 restrictions. Moreover, the fact that the participants were only based at four UK universities could also be a limiting factor for the generalisability of the findings. This emphasises the need for larger and more diverse samples in future studies, and the need for studies following the end of COVID-19 restrictions.

This study has filled a research gap and contributed to our knowledge of the associations between the diet quality and mental health of students during their transition to universities in the UK. The findings can act as a reference for policy makers in terms of designing future university policies relevant to the mental health of students, the diet quality of students, the health education of students, the food security of students and the university food environment. The findings inform the design of relevant studies in the future that may involve clinical samples, and highlight the need for further research including qualitative studies, as well as intervention studies to determine whether interventions to improve diet quality at the university level could reduce mental health issues, and whether interventions to support students under stress may lead to healthier dietary habits.

## Author Contributions

**Conceptualization:** Solomis Solomou, Guillermo Perez-Algorta.

**Data curation:** Solomis Solomou, Guillermo Perez-Algorta.

**Formal analysis:** Solomis Solomou, Heather Robinson, Guillermo Perez-Algorta.

**Investigation:** Solomis Solomou, Guillermo Perez-Algorta.

**Methodology:** Solomis Solomou, Heather Robinson, Guillermo Perez-Algorta.

**Project administration:** Solomis Solomou, Guillermo Perez-Algorta.

**Resources:** Solomis Solomou.

**Software:** Solomis Solomou, Guillermo Perez-Algorta.

**Supervision:** Heather Robinson, Guillermo Perez-Algorta.

**Validation:** Solomis Solomou, Heather Robinson, Guillermo Perez-Algorta.

**Visualization:** Solomis Solomou, Heather Robinson, Guillermo Perez-Algorta.

**Writing – original draft:** Solomis Solomou.

**Writing – review & editing:** Heather Robinson, Guillermo Perez-Algorta.

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
