## [Decision Letter · Decision Letter 0]

14 Aug 2024

PONE-D-24-28220The association of diet quality with the mental health of students during their transition to universityPLOS ONE

Dear Dr. Solomou,

Thank you for submitting your manuscript to PLOS ONE. After careful consideration, we feel that it has merit but does not fully meet PLOS ONE’s publication criteria as it currently stands. Therefore, we invite you to submit a revised version of the manuscript that addresses the points raised during the review process.

We look forward to receiving your revised manuscript.

Kind regards,

Rabie Adel

Academic Editor

PLOS ONE

Journal Requirements:

**Additional Editor Comments:**

Dear Dr. Solomou and Colleagues,

Thank you for submitting your manuscript titled "The association of diet quality with the mental health of students during their transition to university" for consideration. After a careful review, we appreciate the effort that has gone into this work, but we believe that there are few areas where the manuscript could be strengthened. We kindly request that you revise the manuscript to address the reviewers' comments and the following suggestions:

General Comments:

Cross-Sectional Design and Causality:

Comment: The cross-sectional nature of your study limits the ability to draw causal inferences between diet quality and mental health. We recommend adding a more explicit discussion of this limitation in both the abstract and the discussion sections. Additionally, consider suggesting that future research should employ longitudinal designs to better capture the dynamics of these relationships over time.

Action: Please revise the manuscript to clearly articulate this limitation and suggest directions for future research.

Sample Size and Generalizability:

Comment: While your sample size meets the minimum requirement for your primary analyses, it may be insufficient to detect smaller effects, particularly in the moderation and mediation analyses. Furthermore, the study’s focus on students from four UK universities may limit the generalizability of your findings.

Action: We suggest that you address these limitations in the discussion, emphasizing the need for larger and more diverse samples in future studies.

Effect Size and Practical Significance:

Comment: The effect sizes reported in your study are small, which raises questions about their practical significance. It is important to present these findings with an appropriate level of caution.

Action: Please revise the discussion to emphasize the small effect sizes and their implications, ensuring that readers understand the modest nature of these associations.

Overinterpretation of Non-Significant Findings:

Comment: The manuscript discusses non-significant findings from the moderation and mediation analyses at length. While it is important to report these results, care should be taken not to overinterpret them.

Action: We recommend revising these sections to clearly state that these findings are non-significant and to avoid speculative interpretations. Focus on the limitations that may have contributed to these results, such as sample size and variability.

Reliance on Self-Reported Data:

Comment: The study relies on self-reported measures for both diet quality and mental health, which can introduce biases. While self-reports are common in this type of research, it is important to acknowledge the limitations associated with this approach.

Action: Please include a discussion of the potential biases associated with self-reported data and suggest ways that future research could incorporate more objective measures.

Biological Mechanisms Discussion:

Comment: The manuscript speculates about biological mechanisms (e.g., inflammation, HPA axis dysregulation) that might link diet and mental health, but these are not directly measured in the study.

Action: We suggest that you revise this section to clarify that these are hypotheses based on existing literature and not findings from your study. Consider recommending that future studies include biological measures to explore these mechanisms further.

Ethical Considerations:

Comment: While you mention that ethical approval was obtained, it would be beneficial to provide more detail about how participant privacy and data security were protected, particularly given the sensitive nature of mental health data.

Action: Please expand the ethics statement in the methods section to include details on how data confidentiality was maintained and how participant privacy was ensured.

Specific Comments:

Tables:

Comment: There are few issues with the tables, including inconsistent use of symbols for significance levels, lack of units for variables like BMI, and inadequate explanations for certain terms (e.g., sr²). Additionally, the R² and Adjusted R² values are reported without interpretation.

Action: We recommend revising the tables to ensure consistency and clarity. Please provide units where applicable, define all terms in the legend, and offer brief interpretations of the R² and Adjusted R² values.

Introduction: Justification of the Study:

Comment: While the introduction provides a general background on the relationship between diet and mental health, it could benefit from a stronger justification for why this specific study is needed, particularly in the context of UK universities.

Action: Consider expanding the introduction to include a more detailed rationale for the study. Discuss gaps in the existing literature and explain how your study addresses these gaps.

Operational Definitions and Consistency:

Comment: The manuscript should clearly define key terms such as "diet quality," "mental health," and "quality of transition." Consistency in how these terms are used throughout the paper is crucial for clarity.

Action: Review the manuscript to ensure that all key terms are clearly defined at their first mention and used consistently throughout.

Sample Description and Representativeness:

Comment: The description of the sample could be expanded to better highlight its representativeness. How does the demographic profile of your sample compare to the broader population of first-year university students in the UK?

Action: Consider including a brief discussion on the representativeness of your sample in the methods or discussion sections.

Control Variables:

Comment: There may be other potential confounding variables that could influence the relationship between diet and mental health (e.g., socioeconomic status, pre-existing mental health conditions).

Action: Discuss any additional potential confounders that were not controlled for in your analysis. If data are available, consider including these as control variables in your regression models.

Interpretation of Statistical Significance:

Comment: While the manuscript currently focuses on statistical significance, it would be valuable to also discuss the practical significance of the findings.

Action: In the discussion section, provide a more nuanced interpretation of your findings by discussing both statistical and practical significance.

Potential Impact of COVID-19:

Comment: If the study was conducted during or after the COVID-19 pandemic, it’s important to consider how this context might have influenced the findings.

Action: Include a discussion of how the timing of the study (e.g., during or post-COVID-19) might have affected the results.

Conclusion:

We appreciate your hard work and dedication to this important topic and look forward to receiving your revised manuscript.

Thank you for considering these suggestions.

Best regards,

Reviewers' comments:

Reviewer's Responses to Questions

**Comments to the Author**

1. Is the manuscript technically sound, and do the data support the conclusions?

Reviewer #1: Partly

Reviewer #2: Yes

2. Has the statistical analysis been performed appropriately and rigorously? 

Reviewer #1: Yes

Reviewer #2: Yes

3. Have the authors made all data underlying the findings in their manuscript fully available?

Reviewer #1: Yes

Reviewer #2: Yes

4. Is the manuscript presented in an intelligible fashion and written in standard English?

Reviewer #1: Yes

Reviewer #2: Yes

5. Review Comments to the Author

**Reviewer #1:** the comments are included as comment box in the manuscript.

- the sample size is too small. not clear how it was calculated.

- one of the limitations is that the data were self-reported, which increases the bias.

- the assessment tools were not mentioned enough in the abstract.

- not all findings were discussed in the discussion

- a pilot study for assess the reliability was not clear.

- tables need to be rearranged

- the introduction and aims should be rewritten

**Reviewer #2:** I believe it is a very good attempt. The authors have described all the necessary details especially in the methodology section. Results have also been presented effectively. The paper is also focused on something which is worthy of investigation.

6. PLOS authors have the option to publish the peer review history of their article (what does this mean?). If published, this will include your full peer review and any attached files.

Reviewer #1: **Yes: **Buthaina Alkhatib

Reviewer #2: No

---

## [Author Response · Author response to Decision Letter 0]

6 Sep 2024

Please refer to attached rebuttal letter.

---

## [Decision Letter · Decision Letter 1]

2 Oct 2024

The association of diet quality with the mental health of students during their transition to university

PONE-D-24-28220R1

Dear Dr. Solomou,

We’re pleased to inform you that your manuscript has been judged scientifically suitable for publication and will be formally accepted for publication once it meets all outstanding technical requirements.

Kind regards,

Vanessa Carels

Staff Editor

PLOS ONE

Additional Editor Comments (optional):

Reviewers' comments:

Reviewer's Responses to Questions

**Comments to the Author**

1. If the authors have adequately addressed your comments raised in a previous round of review and you feel that this manuscript is now acceptable for publication, you may indicate that here to bypass the “Comments to the Author” section, enter your conflict of interest statement in the “Confidential to Editor” section, and submit your "Accept" recommendation.

Reviewer #1: All comments have been addressed

Reviewer #2: All comments have been addressed

2. Is the manuscript technically sound, and do the data support the conclusions?

Reviewer #1: Yes

Reviewer #2: Yes

3. Has the statistical analysis been performed appropriately and rigorously? 

Reviewer #1: Yes

Reviewer #2: Yes

4. Have the authors made all data underlying the findings in their manuscript fully available?

Reviewer #1: Yes

Reviewer #2: Yes

5. Is the manuscript presented in an intelligible fashion and written in standard English?

Reviewer #1: Yes

Reviewer #2: Yes

6. Review Comments to the Author

Reviewer #1: (No Response)

Reviewer #2: Good attempt. I believe that all the suggestions have now been incorporated into the manuscript and the paper is good to be published.

7. PLOS authors have the option to publish the peer review history of their article (what does this mean?). If published, this will include your full peer review and any attached files.

Reviewer #1: **Yes: **Buthaina Alkhatib

Reviewer #2: No

---

## [Editor Report · Acceptance letter]

7 Oct 2024

PONE-D-24-28220R1 

PLOS ONE

Dear Dr. Solomou, 

I'm pleased to inform you that your manuscript has been deemed suitable for publication in PLOS ONE. Congratulations! Your manuscript is now being handed over to our production team.

Kind regards, 

on behalf of

Dr. Vanessa Carels 

Staff Editor

PLOS ONE